# Methodology for predicting hospital admissions and evaluating recovery rates for coronavirus disease in Japan

**Koichiro Maki** [ID] *

Makisolu LLC., Shiroi-Shi, Chiba, Japan

* makik@pri.nir.jp

## Abstract

In this study, we aimed to propose a method to predict the number of patients needing hospitalization using a combination of available technologies. We developed a method to predict the number of hospital admissions by combining a simple susceptible-infected-recovered (SIR) model with the relationship between the number of new positive cases and the number of hospital admissions, increasing the reliability of each prediction. The accuracy of the concordance between the actual number of patients and the predicted number of hospitalized patients was 99%. Owing to the high accuracy, we were also able to establish a method to evaluate recovery rates. This facilitated determination of the effectiveness of measures implemented throughout Japan to reduce the number of treatment days. The model developed in this study facilitates immediate estimation of the maximum number and timing of hospitalizations based on the peak of new positive cases. Moreover, it provides a statistically true value of the recovery rate required by the mathematical model for investigating countermeasures.

## Introduction

The coronavirus disease (COVID-19) outbreak, which was declared a pandemic by the World Health Organization in March 2020, disrupted healthcare services and presented challenges for several healthcare professionals worldwide [1–4]. In Japan, the insufficiency of hospital beds to accommodate the rapidly increasing number of patients requiring treatment became a pertinent concern [5,6]. To surmount these issues in the future, a predictive model for patient hospitalization is needed. Mathematical models and artificial intelligence have been explored for developing methods to predict the spread of infection [7–9]. However, the existing methods for predicting the spread of COVID-19 have advantages such as good accuracy and good visualization as well as limitations such as the incorporation of incomplete or inaccurate data [10].

**Data availability statement:** The data underlying the results presented in the study are available from Figshare, dx.doi.org/10.6084/m9.figshare.30128617.

**Funding:** The author(s) received no specific funding for this work.

**Competing interests:** The author has declared that no competing interests exist.

Applying predictive models of COVID-19 in clinical practice remains a cumbersome endeavor for medical practitioners [11]. Studies using mathematical models are limited by small sample sizes and insufficient testing of hypotheses, which can lead to overfitting [9,12]. To curtail these drawbacks, samples should be as large as possible, such as a community of several millions of people or more. The utilization of artificial intelligence currently requires the combination of various types of big data with intelligent tools such as machine learning to build predictive models [13]. Predicting the number of infected individuals is challenging with both methods, primarily because several factors are included as parameters for the prediction, and the amount and type of data required increase accordingly.

Forecasting the shortage of beds in a ward entails prediction of the number of hospital admissions, which corresponds to the quarantined persons ($Q$) in the susceptible-infected-quarantined-recovered (SIQR) mathematical model [14]. The function $Q$ is obtained by solving a simultaneous differential equation involving the susceptible ($S$), infected ($I$), and recovered ($R$) variables. However, the function $Q$ has the same issues as those of the mathematical model described above. We proposed that infections can be decomposed by a linear combination of chain reactions [15]. Thus, if the spread of infection occurs within a community where mean-field approximation holds, it can be approximated using the SIR model function. This function can be used to fit the model to a single local peak and predict the future within the period of the peak.

Therefore, this study aimed to devise a method for estimating the number of hospital admissions based on the number of new positive cases. Determining the number of beds needed several days in advance will help maintain a functional and effective medical system.

As a corollary of this objective, we postulated that it would be possible to compare the actual treatment recovery rate of a quarantined person using the estimated recovery rate (0.1) used in the SIR model [15] of community-acquired infection spread. Since the time trend of the average recovery rate for hospitalized patients can be obtained as measured data, the mathematical model can be used to examine various infection control factors that require consideration.

## Materials and methods

### Data sources for the number of daily positive cases and hospital admissions

The open data used in this study included the number of daily positive cases and number of patients requiring treatment published by the Ministry of Health, Labour and Welfare in Japan. The data for daily positive cases are available at https://covid19.mhlw.go.jp/public/opendata/newly_confirmed_cases_daily.csv. The data for the number of people requiring treatment are available at https://covid19.mhlw.go.jp/public/opendata/requiring_inpatient_care_etc_daily.csv. Ethics approval and patient consent were not required as this study dealt only with published data on the number of positives and hospital admissions, and does not contain any personal, identifiable information.

The tabulation of patients requiring treatment was interrupted in September 2022 in some regions; thus, the period of investigation in this study was adjusted to coincide with the valid data range for each prefecture. Data compilation was suspended

on September 26, 2022, in accordance with the government notification regarding 100% testing review [16,17]. We denoted the tabulated data of new positives as $P_i$ $\{i = 1, 2, 3, \cdots\cdots\}$ and the tabulated data of patients requiring treatment as $H_i$ $\{i = 1, 2, 3, \cdots\cdots\}$. $H_i$ includes a few patients with severe illness.

## Conceptual approach to estimating inpatient admissions

To predict the number of individuals requiring treatment, we derived an equation (described in the following section and shown in Fig 1) that correlates the number of new positive cases with the number of hospital admissions. The core procedure involved using the specified formula to convert published values and solutions derived from the mathematical model to prediction results. The inputs for new positives were daily aggregates of $P_i$ and $H_i$ or the solution to a mathematical model that conforms to community-acquired infections. Accordingly, the prediction system consisted of daily data, the solution of the mathematical model, and the prediction tool, as shown in Fig 1. The simplest SIR model with a solution fitted to the infection rate $\beta_0$, recovery rate $\gamma_0$, and community population $N$ was adopted [15]. The prediction tool used two methods. The first entailed conversion of the number of new positives to the number of hospital admissions; the second was used to verify the concordance between the estimations and the ground truth. The actual recovery rate $\gamma$ was verified using daily $H_i$ and $H_i^{cal}$ values converted from the daily $P_i$ data. The new recovery rate obtained after this verification was the reciprocal of the number of treatment days of the quarantined patients. The recovery rate employed in the mathematical models that reproduce the spread of infection is for community-acquired infections, rather than the reciprocal of the number of inpatient treatment days. Nevertheless, various factors that contribute to the variation in the recovery rates of patients who are hospitalized over time can be used in mathematical models to predict the recovery rates of patients with community-acquired infections and in the data analysis [15] of the spread of infection. In that case, $\gamma$ represents feedback for $\gamma_0$ in the mathematical model.

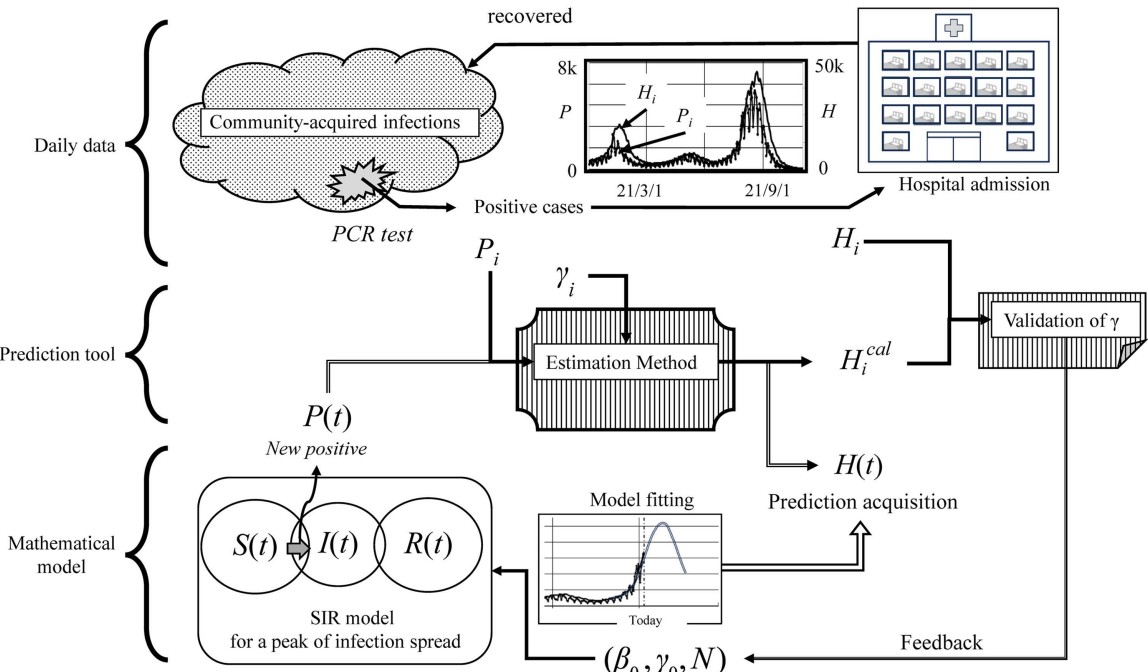

**Fig 1. Diagram of relationships for the prediction.** The prediction is characterized by independent routines for calculating the number of hospital admissions. The system is composed of daily data, prediction tools, and a simple mathematical model.

The recovery rates of individuals are influenced by several factors, including the characteristics of the mutant strain, improvements in hospital treatment techniques, and government guidelines on control measures [18]. Consequently, a small sample leads to substantial uncertainty due to fluctuations, which renders the validation results unreliable. However, the dataset used in this study encompassed the entire Japanese population and was aggregated from all the prefectures. It is the largest such dataset, and all influential factors were averaged out to visualize only nationally representative trends.

The number of hospital admissions was predicted as follows. The solution of the SIR model was obtained by fitting a portion of the peak waveform to the present day. In the model, $\gamma_0$ represented a value known at that time, $\beta_0$ was used as a fitting parameter, and $N$ was an estimated parameter as it provided the peak value of the spread of infection but was indeterminate at the start of the peak rise. The function $P(t)$, which indicates the number of new positives, is typically defined as the daily decrease in the number of susceptible $S(t)$ cases, which was one of the solutions of the SIR model. The estimation of the value and timing of the peak number of hospital admissions was represented by $H(t)$ adjusted for $N$. The value of N was obtained by predicting $P(t)$, and $H(t)$ was transformed from $P(t)$ as follows.

## Mathematical procedures for estimation methods

As shown in Fig 1, patients with infections detected by polymerase chain reaction testing were quarantined in the hospital or home as persons requiring treatment. The number of individuals requiring treatment increased with each new positive case but decreased with recovery after treatment. Accordingly, the differential equation for the number of individuals requiring treatment, $H(t)$, is expressed as follows:

$$\frac{dH}{dt} = P - \gamma H,$$

(1)

where $\gamma$ represents the recovery rate, or more precisely, the treatment completion rate, and its reciprocal is the number of days required for treatment. The solution of this differential equation is equation (2):

$$H(t) = e^{-\int \gamma(t)dt} \left\{ \int_0^t Pe^{\int \gamma(\tau)d\tau} d\tau + H(0) \right\},$$

(2)

which can be expressed as a recurrence formula in equation (3), since $P(t)$ and $\gamma(t)$ have discrete values for each day of unit time:

$$H_{t+1} = P_{t+1} + e^{-\gamma_{t+1}} H_t,$$

(3)

where $H_{t+1}$, $P_{t+1}$, and $\gamma_{t+1}$ refer to $H(t+1))$, $P(t+1)$, and $\gamma(t+1)$, respectively. Iterating equation (3) yields the time variation of the number of people requiring treatment ($H$) from the daily number of people who have tested positive ($P$). The time dependence of $\gamma$ was determined to ensure the consistency of $H_i^{cal}$, with $H_i$ reported by the hospital. $\gamma$ should be a constant or a step function that is constant for an appropriate duration so that it is not affected by daily data fluctuations. The number of admissions in equation (3) can be estimated using discrete values, such as the daily data or a function of the solution of the mathematical model.

## Results

### Calculation of the number of hospitalized patients and estimation of the recovery rate

Fig 2 shows the actual ($H_i$) and calculated ($H_i^{cal}$) number of hospitalized patients in Tokyo from equation (1). The recovery rate $\gamma$ was 0.1 for the entire period, which corresponds to 10 days of treatment [19]. The figure shows several periods where $H_i^{cal}$ deviated from $H_i$. To make them consistent, $\gamma$ was approximated as a step function to make it as insensitive as

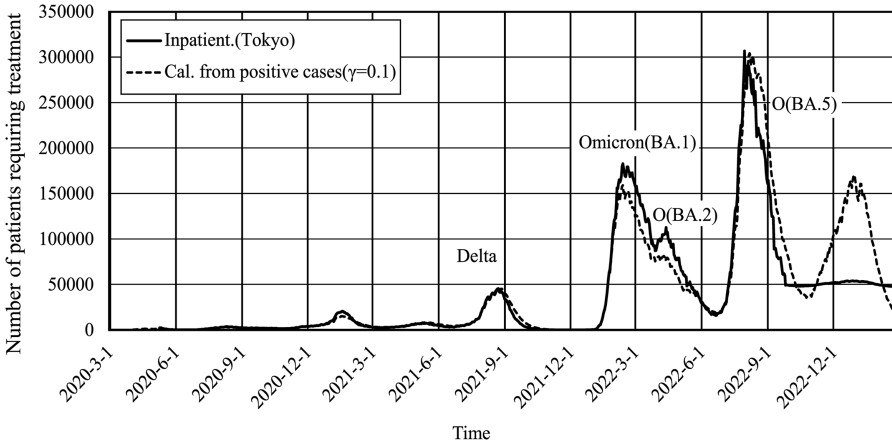

**Fig 2. Number of in-patients and patients requiring treatment calculated from the new positive cases in Tokyo at a recovery rate of γ = 0.1.** The solid line shows the data for inpatients in Tokyo, and the dashed line shows the calculated number of inpatients.

possible to the fluctuations in daily data. The recovery rate of 0.1 was adequate for the period through December 2021, excluding the periods when variants emerged. These are the periods corresponding to the Alpha and Delta variants. The Omicron variant, including the BA.1, BA.2, and BA.5 sub-variants, emerged after January 2022. The same recovery rate of 0.1 was used in the mathematical model for community-acquired infections. This indicates that the mean hospital recovery rate can be employed in the actuarial model as the recovery rate for community-acquired infections. It was difficult to confirm $\gamma = 0.1$ for the Omicron period. The genome was identified by the National Institute of Infectious Diseases (NIID) in Japan [20].

The upper graph in Fig 3 shows the step function of $\gamma$, and the lower graph shows both curves and their overlap. Values of $\gamma$ less than 0.1 indicated more days of treatment, which was observed for two events of worsening infections. The first lasted from January to February 2021, while the second lasted from February to May 2022; $\gamma$ equaled to 0.08 in both instances. Conversely, $\gamma$ values greater than 0.1 indicated fewer days of treatment, which were observed during two periods of declining infections. The first period lasted from October to November 2021 ($\gamma = 0.13$), while the second lasted from August to September 2022 ($\gamma = 0.14$). The concordance between the number of reported persons requiring treatment and the number of hospital admissions estimated from the reported new positive cases was approximately 99% or better. The coefficient of determination between the two was $R^2 = 0.993$. The test period was 985 days, from January 16, 2020, when the data became available, to September 26, 2022.

## Recovery rates for all prefectures in Japan

A status diagram of $\gamma$ values for all prefectures in Japan is shown in Fig 4, to facilitate the visualization of the systematic changes with common factors. In terms of the national trends of recovery rates, $\gamma$ was greater than 0.1 for two periods, which is the same as the result for Tokyo. In contrast, $\gamma$ was not less than 0.1 for any period or prefecture, and the national trend was not necessarily the same as that of Tokyo. Therefore, the low values of $\gamma$, indicating that patients required longer treatment during a surge in hospital admissions, can be attributed to the treatment process and environment at hospitals in each region. It is difficult to determine the cause of a phenomenon that is not common throughout the country because it requires meticulous investigation using detailed data. The two periods with larger $\gamma$, corresponding to fewer days of treatment, are roughly estimated to be September–November 2021 and July–October 2022, as indicated by $\gamma_{larger-ex}$. In July–August 2022, several prefectures had overlaps of the two states of $\gamma$ and $\gamma_{larger-ex}$ due to a steep surge in hospital admissions.

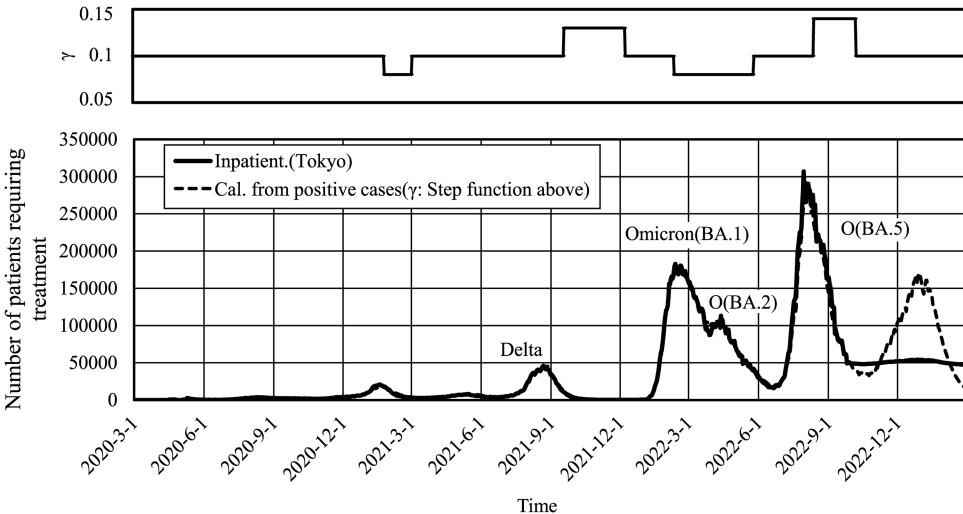

**Fig 3. Number of inpatients and patients requiring treatment calculated from the number of new positive cases in Tokyo by introducing the recovery rate of the step function.** The step function for γ is shown at the top of the figure. The solid line shows the data for inpatients in Tokyo, and the dashed line shows the calculated number of inpatients.

## Reflecting recovery rates in mathematical models

The $\gamma_{larger-ex}$ for the two periods highlighted in the previous section can be considered as the recovery rate of community-acquired infections, since the same value, irrespective of prefecture, implies a universal phenomenon. As the recovery rate increases, the infection rate apparently declines. Studies using the equivalent SIR model of the chain reaction [15,19], which closely reproduces the Tokyo data, have reported an anomaly, i.e., a sudden reduction in the infection rate during the same two periods as $\gamma_{larger-ex}$. Reflecting the increased recovery rate of $\gamma_{larger-ex}$ during these periods resolved this anomaly in the new positive case data proportional to community-acquired infections.

These two periods of greater $\gamma$ values were close to the timing of the nationwide vaccinations [21,22]. Specifically, the two periods correspond to the second half of the peak of the Delta mutant infection and the final tail of the BA.2 mutant infection. Additional statistical verification is required before establishing definitive causality between the recovery rate and vaccination.

## Architecture for estimating the maximum number of inpatients from the daily positive cases

After the appropriate recovery rate for community-acquired infections was determined, the SIR model was used to predict future trends. The objective of the prediction method was to estimate the number of hospitalized patients in advance as much as possible at the time of infection spread. This method could not predict future outbreaks. Given that the initial components of the SIR model function can be approximated using an exponential function, the infection rate $\beta_0$ can be determined by fitting a linear upward slope to the daily data of the $P_i$ curve, as illustrated in the logarithmic representation in Fig 5a. The upward slope was $\approx \beta_0 - \gamma_0$, which did not permit determination of $N$ with certainty. Consequently, we estimated $N$ by applying the principle that the peak of new positives is contingent upon the size of the basic community of infection. Fig 5b shows the logarithmic representation of the function $P(t)$ with varying infection rate $\beta_0$ of 0.5, 0.4, 0.3, 0.2, 0.15, and 0.12. The initial condition was that there had to be one infected person at time $t=0$.

The value of $N$ was determined when the data of new positives reached a maximum. At this time, the relationship between the maximum $P(t)$ and $H(t)$ based on equation (2) was as follows: $H(t)$ was approximately $\frac{1}{\gamma_0}$ multiplied by $P(t)$, and $H(t)$ reached its maximum value approximately $\frac{1}{\gamma_0}$ days after $P(t)$. These formulae were used to calculate the peak

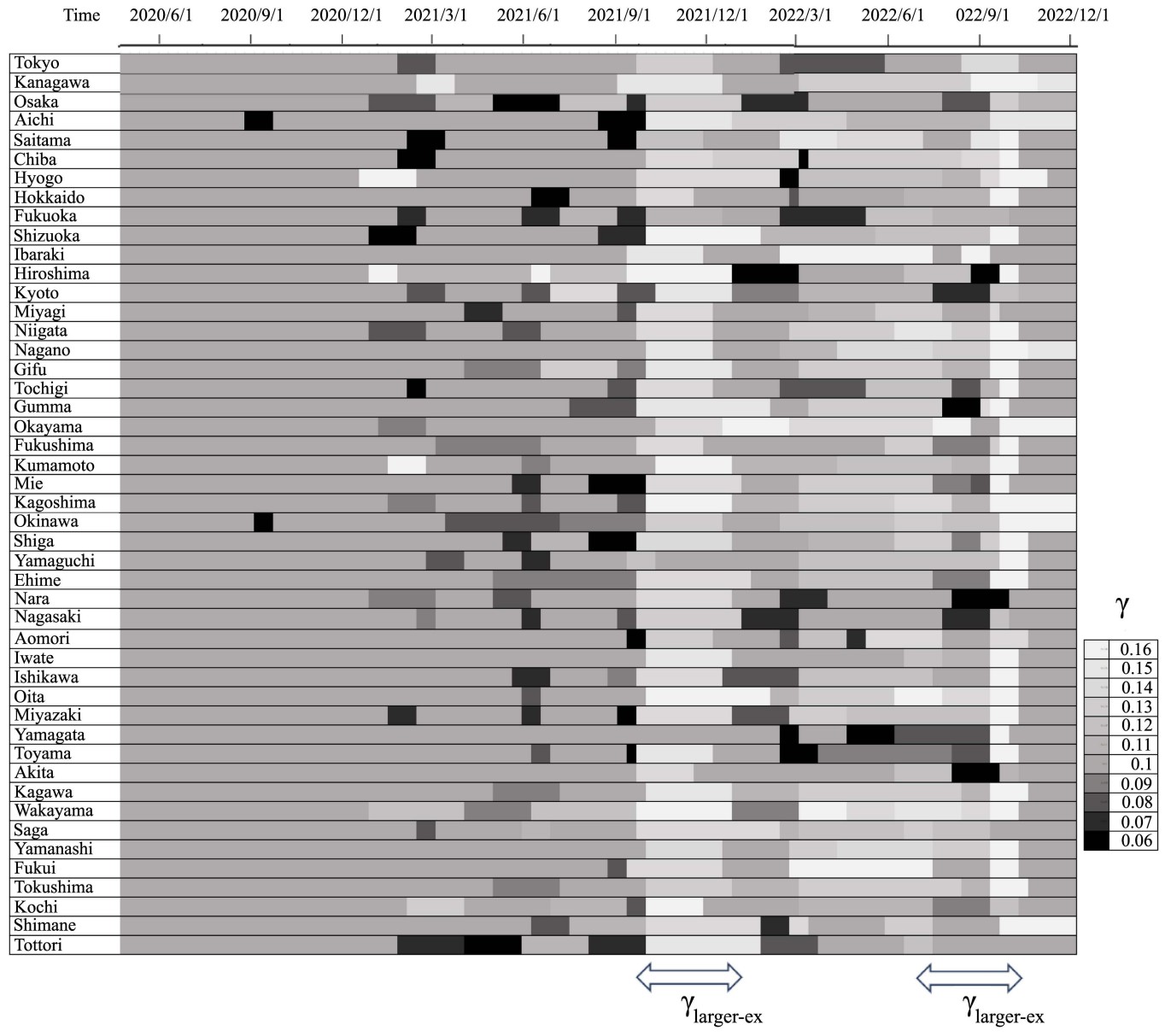

**Fig 4. The γ distribution for each prefecture in Japan.** Two periods with large recovery rates γlarge-ex were identified. The vertical axis shows the prefectures in order of the largest population from top to bottom, and the horizontal axis shows time on the same scale as that shown in Fig 2. To convey whether γ is greater or smaller than 0.1, the figure is color-coded in black and gray, respectively.

of patient hospitalizations as soon as the peak of new positive results could be estimated. Fig 6a shows the relationship between the maximum number of patients requiring treatment as $H_{max}$ and the time to reach it as $t_H$, as well as that between the maximum number of new positive results as $P_{max}$ and the time to reach it as $t_P$. In more precise terms, the dependence of the maximum values of $P$ and $H$ on the infection rate $\beta_0$ was calculated from equation (2), as illustrated in Fig 6b. Consequently, the maximum number of hospitalized patients was reached sooner as the infection rate increased. An increase in $\beta_0$ from 0.1 to 0.3 indicated that the maximum number of hospitalizations ($H_{max}/P_{max}$) declined from 10 to 8

a)

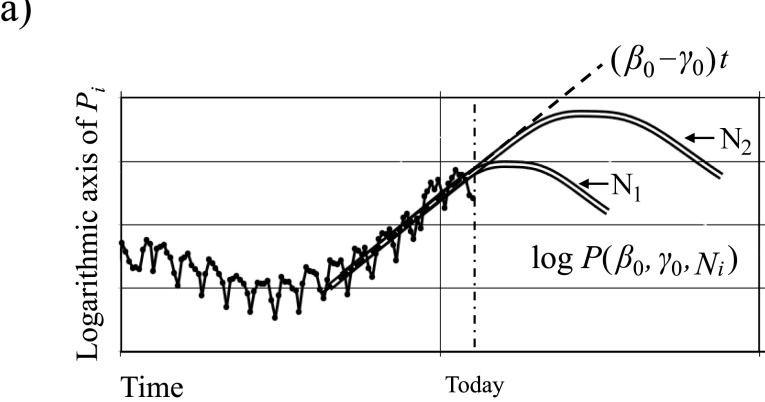

b)

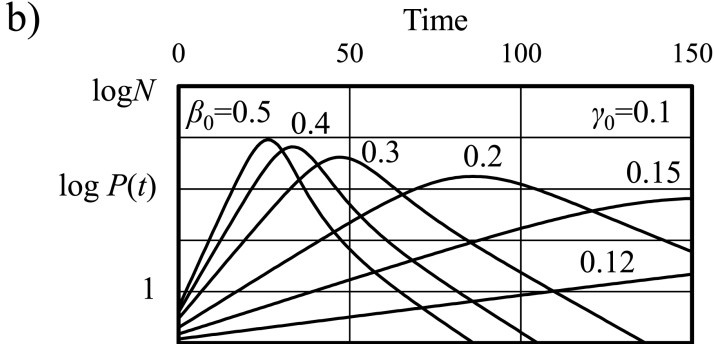

**Fig 5. Prediction procedure for new positive cases. a)** Estimation method with a single peak. **b)** Logarithmic representation of function *P* with varying infection rate β. *N* denotes the population of the community.

multiplied by the maximum number of new positive cases, and the difference in the number of days between the two maximums $(t_H - t_P)$ declined from 10 to 6, for both cases in which the value of $\gamma_0$ was 0.1.

## Discussion

In this study, we devised a method for predicting the number of hospital admissions based on the number of new positive cases. This method facilitated estimation of the number of individuals who would be quarantined using a simple SIR model. In addition, comparison of the predictions based on the number of new positive results and number of persons requiring treatment revealed variations in the number of days required for treatment. The recovery rate for community-acquired infections was derived from the number of treatment days required and can be incorporated into the mathematical model. This cyclical system can serve as a tool for forecasting the risk of hospital bed insufficiency. Concurrently, the temporal variation in the recovery rate can be used to discern alterations in the characteristics of the mutant strains and fluctuations in the efficacy of treatment in the hospital and national preventive measures.

Vaccination was the most probable cause of the increased recovery rates observed during the spread of the Delta strain, despite its tendency to cause severe disease [23]. The second increase in the recovery rate during the BA.5 mutant period was $\gamma = 0.14$, which is slightly higher than the recovery rate $\gamma = 0.13$ during the period of predominance of the Delta mutant. This may be attributed to the overlap of the higher recovery rates over a longer period since March 2022. This is consistent with reports that the Omicron strain is less likely to cause severe disease [24]. $\gamma_{larger-ex}$ is also not inconsistent

a)

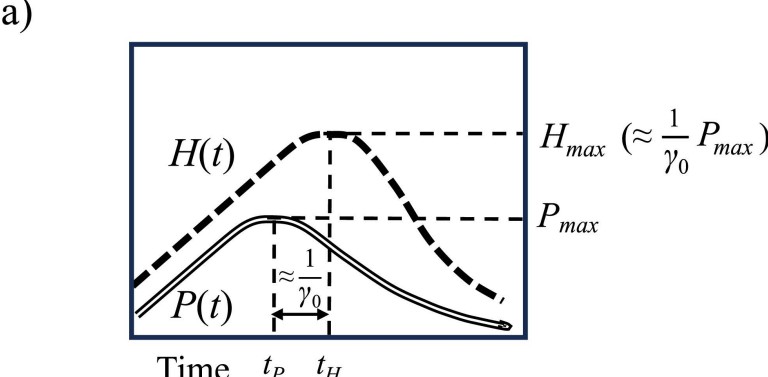

b)

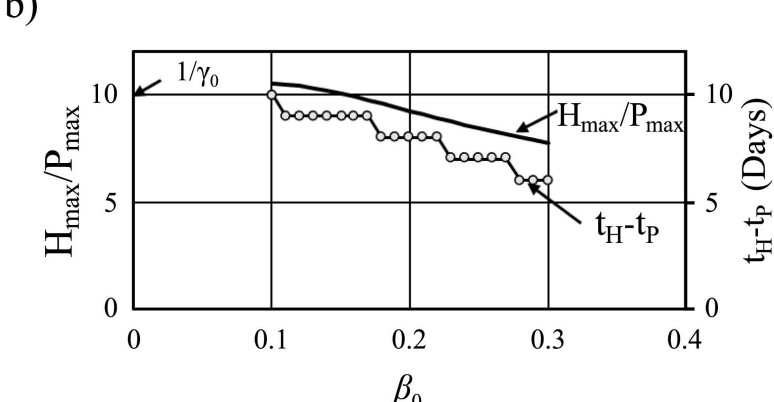

**Fig 6. Prediction procedure for number of hospitalized patients. a)** Relationship between the peak values of $P$ and $H$. **b)** Dependence of the relationship between both peak values on the infection rate β. $H_{max}$ represents the maximum number of admissions and $P_{max}$ is the maximum number of new positives.

with reports of the high vaccine efficacy for up to less than 4 months [25,26]. Conversely, there were periods where the recovery rate $\gamma$ was below 0.1, as shown in Figs 3 and 4. Since this value was not the same for all prefectures, it can be assumed that it depends on the situation of the region and hospitals, as mentioned above. The recovery rate could be determined for each hospital, and the results would be useful in initiating supportive measures for the hospital.

In light of these findings, this study focused on verifying whether the number of new positives and the number of hospital admissions can be used to accurately predict the recovery rate. According to previous studies, prediction of the number of hospital admissions using the SIQR model corresponds to the number of patients under quarantine [27–29]. Their main focus was investigating the formulation and solution of coupled differential equations, and their scope did not extend to the reproduction of the number of hospital admissions over a long period. Given these circumstances, it would rather be beneficial to apply the recovery rates determined by this method to mathematical models such as SIR, SIQR, and SEIR, which require the recovery rate parameter. Equation (1) is a "conservation equation" for the number of hospitalized patients. This relationship should hold true for any closed system, irrespective of the complexity of the model employed.

A limitation of this method is that the scenario on which the relationship in equation (3) and this estimation method are based is no longer valid. In other words, all new positive cases are no longer managed in quarantine as (persons) needing treatment. In such circumstances, it is necessary to model and validate the relationship on which the estimation method is

based when predicting hospital admissions, considering the conditions of the newly admitted patients (e.g., the severity of illness).

However, applying this to each hospital is expected to reveal differences in factors specific to each hospital, such as disease severity, age, and comorbidities of the patients, and the healthcare system. If the data collection system utilized in this study can be established as a countermeasure in the next pandemic, it could be widely adopted internationally to determine recovery rates in mathematical models and predict bed shortages.

## Conclusions

To address the risks of insufficient hospital beds, it is important not only to predict hospital admissions but also to leverage artificial intelligence tools to address the needs of patients [30] according to their medical conditions, in the event of a surge in hospital admissions. It is hoped that these efforts will curtail human intervention in the medical field. To mitigate the risk of a shortage of hospital beds during a pandemic emergency, it is imperative that strategies such as tools to predict the number of patients requiring treatment within a few days in response to trends in new positive cases and measures to secure sickbeds be prepared on a global scale.

## Acknowledgments

I would like to thank Editage (www.editage.com) for English language editing.

## Author contributions

**Conceptualization:** Koichiro Maki.

**Data curation:** Koichiro Maki.

**Formal analysis:** Koichiro Maki.

**Investigation:** Koichiro Maki.

**Methodology:** Koichiro Maki.

**Validation:** Koichiro Maki.

**Visualization:** Koichiro Maki.

**Writing – original draft:** Koichiro Maki.

**Writing – review & editing:** Koichiro Maki.

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
