## [Decision Letter · Decision Letter 0]

26 Aug 2025

Dear Dr. Maki,

We look forward to receiving your revised manuscript.

Kind regards,

Etsuro Ito, Ph.D.

Academic Editor

PLOS ONE

Journal Requirements:

Reviewers' comments:

Reviewer's Responses to Questions

**Comments to the Author**

1. Is the manuscript technically sound, and do the data support the conclusions?

Reviewer #1: Yes

Reviewer #2: Yes

2. Has the statistical analysis been performed appropriately and rigorously?

Reviewer #1: No

Reviewer #2: Yes

3. Have the authors made all data underlying the findings in their manuscript fully available?

Reviewer #1: Yes

Reviewer #2: Yes

4. Is the manuscript presented in an intelligible fashion and written in standard English?

Reviewer #1: Yes

Reviewer #2: Yes

Reviewer #1: The manuscript presents a methodology for predicting hospital admissions for COVID-19 in Japan by integrating a simple SIR model with observed relationships between new positive cases and hospitalization data. The work also extends the approach to estimate recovery rates, enabling retrospective evaluation of medical interventions and policy measures. The topic is timely and relevant, and the dataset used is comprehensive, covering the entire Japanese population. The manuscript is generally well-structured and clearly written, with a logical flow from problem statement to methodology, results, and implications but needs to address the below issues.

1. While the model performs well with Japanese national data, it is unclear how adaptable the methodology is to regions with different healthcare systems, testing policies, or hospitalization criteria. A brief discussion of applicability to other contexts would improve the manuscript’s impact.

2. The approach assumes that all new positive cases requiring treatment are tracked in the dataset. As acknowledged in the limitations, this assumption no longer holds in many contexts. It would be useful to explore adjustments or correction factors for partial case tracking.

3. The model’s predictive performance is reported primarily in terms of concordance rates. However, there is no formal uncertainty analysis (e.g., confidence intervals, sensitivity analysis) to assess robustness under varying data quality or parameter uncertainty.

4. While the recovery rate (γ) is used effectively for retrospective evaluation, the causal interpretation—particularly its linkage to vaccination campaigns—relies on temporal coincidence. Additional statistical testing or correlation analysis would strengthen these claims.

Reviewer #2: The manuscript addresses a public health challenge, i.e. predicting hospital admissions and recovery rates during the COVID-19 pandemic in Japan. The approach combines a simple SIR model with empirical hospitalization data. The findings, particularly the ability to estimate recovery rates and hospitalization peaks, are relevant for pandemic preparedness and healthcare resource planning.

However, the manuscript requires major revisions.

1. The main claimed novelty is the integration of hospitalization data to refine recovery rates. The manuscript should more clearly articulate what distinguishes this work from existing SIQR or SEIR modeling studies.

2. The discussion should elaborate on how this methodology could be generalized beyond COVID-19 or Japan, which would strengthen its broader applicability.

3. The prediction accuracy of “99% concordance” is mentioned several times. This figure is very high and requires a clear explanation of how it was calculated, i.e. R², mean absolute percentage error or concordance correlation coefficient.

4. The paper should provide sensitivity analyses showing how results change with different assumptions about γ.

5. The validation is based solely on retrospective Japanese data. To strengthen the case for the model, the author should attempt an out-of-sample validation, i.e. training on data from one wave, testing on another or using one prefecture as training and another for validation.

6. Comparison with more complex models, i.e. machine learning based predictions or SEIR models, would help demonstrate advantages and trade-offs of this simpler approach.

7. The manuscript equates recovery rates derived from hospitalization data with those of community-acquired infection models. This assumption might not be fully justified since hospitalized patients may differ systematically from the broader infected population. The author should discuss this.

8. The attribution of higher recovery rates to vaccination campaigns is plausible but speculative. The discussion should be more cautious and reference direct evidence linking vaccination timing to observed recovery rates.

9. While some limitations are acknowledged others are underdeveloped. Additional limitations that should be explicitly discussed include: Exclusion of age, comorbidities and regional healthcare capacity; reliance on reported “positives” which may not reflect true infection rates due to testing capacity and policy changes; assumption that all positive cases initially require treatment / quarantine which is no longer accurate.

10. Some references are heavily Japan centric. Including more international studies on hospitalization prediction would situate the work broadly in the literature.

**Do you want your identity to be public for this peer review?** For information about this choice, including consent withdrawal, please see our Privacy Policy

Reviewer #1: No

Reviewer #2: No

---

## [Decision Letter · Decision Letter 1]

30 Sep 2025

Methodology for predicting hospital admissions and evaluating recovery rates for coronavirus disease in Japan

PONE-D-25-39893R1

Dear Dr. Maki,

We’re pleased to inform you that your manuscript has been judged scientifically suitable for publication and will be formally accepted for publication once it meets all outstanding technical requirements.

Kind regards,

Etsuro Ito, Ph.D.

Academic Editor

PLOS ONE

Reviewers' comments:

Reviewer's Responses to Questions

**Comments to the Author**

Reviewer #2: All comments have been addressed

2. Is the manuscript technically sound, and do the data support the conclusions?

Reviewer #2: Yes

3. Has the statistical analysis been performed appropriately and rigorously?

Reviewer #2: Yes

4. Have the authors made all data underlying the findings in their manuscript fully available?

Reviewer #2: Yes

5. Is the manuscript presented in an intelligible fashion and written in standard English?

Reviewer #2: Yes

Reviewer #2: I recommend publication as is. All revisions were performed as suggested. Congratulations to the authors.

**Do you want your identity to be public for this peer review?** For information about this choice, including consent withdrawal, please see our Privacy Policy

Reviewer #2: No

---

## [Editor Report · Acceptance letter]

PONE-D-25-39893R1

PLOS ONE

Dear Dr. Maki,

I'm pleased to inform you that your manuscript has been deemed suitable for publication in PLOS ONE. Congratulations! Your manuscript is now being handed over to our production team.

Kind regards,

on behalf of

Prof. Etsuro Ito

Academic Editor

PLOS ONE